# Influence of Phenolic-Food Matrix Interactions on *In Vitro* Bioaccessibility of Selected Phenolic Compounds and Nutrients Digestibility in Fortified White Bean Paste

**DOI:** 10.3390/antiox10111825

**Published:** 2021-11-18

**Authors:** Łukasz Sęczyk, Urszula Gawlik-Dziki, Michał Świeca

**Affiliations:** 1Department of Industrial and Medicinal Plants, University of Life Sciences in Lublin, 15 Akademicka Street, 20-704 Lublin, Poland; 2Department of Biochemistry and Food Chemistry, University of Life Sciences in Lublin, 8 Skromna Street, 20-704 Lublin, Poland; urszula.gawlik@up.lublin.pl

**Keywords:** phenolic compounds, food matrix, interactions, *in vitro* digestion, bioaccessibility, digestibility, antioxidant activity, white bean, protein, starch

## Abstract

This model study aimed to evaluate the effect of phenolic–food matrix interactions on the *in vitro* bioaccessibility and antioxidant activity of selected phenolic compounds (gallic acid, ferulic acid, chlorogenic acid, quercetin, apigenin, and catechin) as well as protein and starch digestibility in fortified white bean paste. The magnitude of food matrix effects on phenolics bioaccessibility and antioxidant activity was estimated based on “predicted values” and “combination indexes”. Furthermore, the protein–phenolics interactions were investigated using electrophoretic and chromatographic techniques. The results demonstrated phenolic–food matrix interactions, in most cases, negatively affected the *in vitro* bioaccessibility and antioxidant activity of phenolic compounds as well as nutrient digestibility. The lowest *in vitro* bioaccessibility of phenolic compounds in fortified paste was found for quercetin (45.4%). The most negative impact on the total starch digestibility and relative digestibility of proteins was observed for catechin–digestibility lower by 14.8%, and 21.3% (compared with control), respectively. The observed phenolic–food matrix interactions were strictly dependent on the applied phenolic compound, which indicates the complex nature of interactions and individual affinity of phenolic compounds to food matrix components. In conclusion, phenolic–food matrix interactions are an important factor affecting the nutraceutical and nutritional potential of fortified products.

## 1. Introduction

In recent years phenolic compounds have attracted considerable interest as functional supplements of various food products due to their promising physiological activities. Numerous *in vitro* and in vivo studies have shown that polyphenols are natural agents with the potential for preventing degenerative and chronic diseases related to oxidative stress such as cardiovascular diseases, cancer, diabetes mellitus, and neurodegenerative diseases. Furthermore, these compounds affect the important functional characteristics of food products, including colour, flavour, odour, and oxidative stability, among others [1,2,3].

Because of natural occurrence, safety and bioactivity, these compounds are commonly used for food fortification [4]. Food fortification is defined as the addition of exogenous substances to basic food products to improve their pro-health properties [5]. It is a common way for improving the nutraceutical quality of food; however, some previous studies have shown that incorporation of phenolic compounds into food systems resulted in significant changes in the nutritional and nutraceutical quality as a consequence of phenolic interactions with food matrix components [6,7].

It was reported that phenolic compounds can interact with macromolecules such as proteins, carbohydrates, and lipids [8,9,10,11]. Taking into account the nutraceutical and nutritional quality of products, interactions of phenolic compounds with food components can be considered in two main aspects. Firstly, the binding of polyphenols by food matrix can decrease their “free” level and further bioaccessibility defined as a part of the ingested substance that is potentially available for absorption in the digestive tract [2,12,13,14,15]. Secondly, interactions can cause physicochemical and structural changes of nutrients affecting their digestibility [8,9,14,16,17]. Both, these aspects are crucial for the bioavailability and bioactivity of polyphenols.

Due to limited information in the literature about the phenolic–food matrix interactions and their consequences, regarding food fortification, the objective of this study was to evaluate the food matrix effects on the *in vitro* bioaccessibility of selected phenolic compounds as well as nutrients’ digestibility. For this purpose selected phenolic compounds (gallic acid, ferulic acid, chlorogenic acid, quercetin, apigenin, and catechin) were incorporated into the white bean paste–a model low processed food product. *In vitro* digestion procedure was applied to evaluate phenolics bioaccessibility and protein and starch digestibility. Phenolic content and antioxidant activity based on ABTS radical cation decolourization assay, ferric (III) reducing antioxidant power, ferric (II) chelating power, and hydroxyl radical scavenging activity, for samples before and after simulated digestion, were also determined. To show the magnitude of the food matrix effects on the above–mentioned properties “predicted values” and “combination indexes” were estimated. Furthermore, the protein–phenolics interactions were investigated using electrophoretic and chromatographic techniques.

## 2. Materials and Methods

### 2.1. Chemicals

Phenolic standards: gallic acid (GA), ferulic acid (FA), chlorogenic acid (3-*O*-caffeoylquinic acid; CGA), quercetin (Q), and (+)-catechin (CAT) were purchased in Sigma–Aldrich company (St. Louis, MO, USA); apigenin (A) was purchased from the Roth company (Karlsruhe, Germany). Components of the simulated digestion fluids: α-amylase from hog pancreas (50 U/mg; EC 3.2.1.1), pepsin from porcine gastric mucosa (250 U/mg; EC 3.4.23.1), pancreatin from porcine pancreas (4× USP specifications), bile extract porcine and reagents for antioxidant analysis: 2,2′-azino-bis (3-ethylbenzothiazoline-6-sulfonic (ABTS), potassium ferricyanide, ferrozine, sodium salicylate were obtained from Sigma–Aldrich Company (St. Louis, MO, USA). HPLC grade reagents–acetonitrile and formic acid were purchased from Merck (Darmstadt, Germany). All other reagents were of analytical grade.

### 2.2. Plant Material

Commercially available white kidney bean (*Phaseolus vulgaris* (L.) var. Jas Karlowy) was purchased from a local store. White kidney bean seeds were manually dehulled and milled into the flour using a laboratory mill (IKA M20, IKA Werke GmbH & Co. KG, Staufen, Germany). Powdered plant material was sieved through the 0.50 mm square holes screen, sealed in plastic containers, and stored at −20 °C until analysis in a laboratory freezer (LFE 700, Arctico, Esbjerg, Denmark).

### 2.3. White Bean Paste Preparation

White bean flour was suspended in distilled water with a ratio of 1:4 (*w*/*w*), incubated in a water bath (LWT 2/150, WSL Sp. z o.o., Swietochlowice, Poland) at 100 °C for 30 min, mixing every 5 min. After cooling down to room temperature, the amount of water evaporated during the cooking process was made up to the initial amount (based on the weight of white bean suspension before and after hydrothermal treatment). Then, white bean paste was mixed with solutions of analyzed phenolic compounds (GA, FA, CGA, Q, CAT, A). Each selected phenolic compound was incorporated in the amount of 10 mg per 5 g of hydrated white bean paste (corresponded to 1 g of dry-weight white bean paste). In a control sample (C) solution of the phenolic compound was replaced by distilled water. To exclude the effect of white bean paste preparation and further processing on the stability of phenolic compounds, samples containing solutions of polyphenols without bean paste were prepared. Phenolic samples without the food matrix, prepared in the same way, were also used for the determination of the experimental amount (EV) of phenolic compounds (see Section 2.10). All samples were frozen and freeze-dried in a laboratory freeze-drier (Labconco FreeZone, Kansas City, MO, USA).

One part of the samples was treated with distilled water, centrifuged (MPW 350R, MPW Med. Instruments, Warsaw, Poland) for 15 min (9000× *g*), and the supernatants were filtered using an ultrafiltration membrane (Amicon, Merck Millipore, Darmstadt, Germany) with a 5 kDa molecular weight limit (samples before *in vitro* digestion–BD) and the second was subjected to the in *vitro* gastrointestinal digestion (samples after *in vitro* digestion–AD).

### 2.4. In Vitro Digestion Procedure

Simulated digestion, including three main consecutive steps: oral, gastric, and intestinal phase, performed according to the procedure described by Minekus et al. [12], with a small modification. Before the initial step of digestion, 2.5 g of freeze-dried samples were rehydrated with 2.5 mL of distilled water to obtain pasta-like consistency at the oral phase of digestion, as suggested in the protocol [12]. To unify the procedure for the phenolic samples without food matrix (containing 25 mg of polyphenols) were also hydrated with 2.5 mL of distilled water.

For the oral digestion, hydrated samples were homogenized in the ratio 1:1 with simulated salivary fluid (SSF) consisting of 3.5 mL of SSF electrolyte stock solution (ESS), 0.975 mL of distilled water, 0.025 mL 0.3 M CaCl_2_ and 0.5 mL salivary α–amylase solution (1500 U mL^−1^; prepared in SSFESS). The pH was checked and, when required, adjusted with 1 M NaOH or 1 M HCl to pH 7. The samples were incubated in the dark with continuous shaking for 2 min at 37 °C.

Samples after the oral phase were mixed in the ratio 1:1 with simulated gastric fluid (SGF) consisting of 7.5 mL of SGF electrolyte stock solution, 0.695 mL of distilled water, 0.005 mL of 0.3 M CaCl_2_, 0.2 mL of 1 M HCl and 1.6 mL of pepsin (25,000 U mL^−1^ made up in SGFESS). The pH was checked and, when required, adjusted with 1 M NaOH or 1 M HCl to pH 3. The samples were incubated in the dark with continuous shaking for 2 h at 37 °C.

Gastric bolus was combined in the ratio 1:1 with simulated intestinal fluid (SIF) consisting of 11 mL simulated intestinal fluid electrolyte stock solution (SIFESS), 1.31 mL of distilled water, 2.5 mL fresh bile (160 mM), 0.04 mL of 0.3 M CaCl_2_, 0.15 mL of 1 M NaOH, and 5.0 mL of a pancreatin solution (800 U mL^−1^ made up in SIFESS). The pH was checked and, when required, adjusted with 1 M NaOH or 1 M HCl to pH 7. The samples were incubated in the dark with continuous shaking for 2 h at 37 °C.

SSFESS, SGFESS, SIFESS were prepared strictly according to the procedure described by Minekus et al. (2014) [12].

After the simulated digestion, the samples for phenolic content and antioxidant analysis were centrifuged (MPW 350R, MPW Med. Instruments, Warsaw, Poland) for 15 min, 9000× *g* and the supernatants were filtered using an ultrafiltration membrane (Merck Millipore, Darmstadt, Germany) with a 5-kDa molecular weight limit to obtain potentially bioaccessible fractions. Parallelly, in the same way, blank samples for further analysis containing only digestive components and electrolytes (without food matrix or phenolics) were prepared.

### 2.5. Quantitative Analysis of Phenolic Compounds by High–Performance Liquid Chromatography (HPLC)

Chromatographic analysis of phenolic compounds in the BD and AD samples was carried out according to Sęczyk et al. [18]. Samples were analyzed with a Varian ProStar HPLC system (Varian, Palo Alto, CA, USA) equipped with a ProStar diode array detector (DAD). The analytical column was a 250 mm × 4.6 mm COSMOSIL 5C18-MS-II Packed Column (Nacalai Tesque, Inc., Kyoto, Japan). The mobile phase consisted of 0.1% formic acid in water (solvent A) and 0.1% formic acid in acetonitrile (solvent B). Elution mode was programmed as follows: 0–5 min, 5% B; 5–20 min, 5–100% B; 20–25 min, 100% B; 25–30 min 100–5% B. The flow rate was 1 mL/min and the column thermostat was set at 30 °C. The spectrophotometric detection was performed at wavelengths: 270 nm for GA, 280 nm for CAT, 325 nm for FA and CGA, and 350 nm for Q and A. Quantification of phenolic compounds was carried out using external standard calibration curves.

### 2.6. Determination of Antioxidant Activity

Antioxidant activity was evaluated based on ABTS radical cation decolourization assay (ABTS) [19], ferric (III) reducing antioxidant power (RP) [20], ferric (II) chelating power (CHEL) [21] and hydroxyl radical scavenging activity (^•^OH) [22]. The measurements were conducted on the UV-Vis spectrophotometer (UV 1601PC, Shimadzu Co., Ltd., Kyoto, Japan). Results for ABTS, RP, and ^•^OH were expressed as Trolox equivalents (TE) in mg per 1 g dry weight (DW) of bean paste. Results for CHEL were expressed as ethylenediaminetetraacetic acid equivalents in mg (EDTA) per 1 g DW of bean paste. 

### 2.7. In Vitro Bioaccessibility

The percentage of *in vitro* bioaccessibility was estimated according to the following equation:BA (%) = (AD/BD) × 100%(1)
where: BA—bioaccessibility (%); AD—phenolics content or antioxidant activity of samples after simulated digestion; BD—phenolics content or antioxidant activity of samples before simulated digestion.

### 2.8. Nutrients Analysis

#### 2.8.1. Total Dietary Fiber Content

The total dietary fiber (TDF) content was determined by the gravimetric method using the commercial Total Dietary Fiber Kit (Megazyme, Ireland) following the manufacturer’s procedure [23]. Results were expressed in mg per 1 g DW of white bean paste.

#### 2.8.2. Total Starch Content

The total starch (TS) content was determined using Total Starch Assay Kit (amyloglucosidase/α–amylase method) suitable for food products (Megazyme, Ireland) according to manufacturer’s procedure [24], with slight modification–glucose content was determined based on DNS assay [25] using UV-Vis spectrophotometer (UV 1601PC, Shimadzu Co., Ltd., Kyoto, Japan). Starch was calculated as glucose × 0.9. Results were expressed in mg per 1 g DW of white bean paste.

#### 2.8.3. Available Starch Content

The potentially available starch (AS) content was determined based on the above described *in vitro* digestion procedure [12], with slight modification–*in vitro* digestion procedure was performed for the less initial amount of digested material (1 g of freeze-dried WBP) and using adequately fewer reagents. After, *in vitro* digestion samples were centrifuged (MPW 350R, MPW Med. Instruments, Warsaw, Poland) for 10 min, 5000× *g* at 20 °C to obtain supernatants. Supernatants were mixed with a ratio of 1:4 with glucoamylase (33 U/mL; Megazyme, Ireland) in sodium acetate buffer pH 5, and incubated for 30 min at 50 °C. Glucose content was determined based on a DNS assay [25] using a UV-Vis spectrophotometer (UV 1601PC, Shimadzu Co., Ltd., Kyoto, Japan). Available starch was calculated as glucose × 0.9. Results were expressed in mg per 1 g (dry weight) of white bean paste.

#### 2.8.4. Starch Digestibility

The starch digestibility (STD) was evaluated based on TS content and AS content according to the following equation:STD (%) = AS/TS × 100%(2)
where: STD–starch digestibility (%), AS—available starch, TS—total starch.

#### 2.8.5. Protein Digestibility

The protein digestibility was determined based on the above described *in vitro* digestion procedure [12], with slight modification—the experiment was performed for the less initial amount of digested material (1 g of freeze-dried WBP) and using adequately fewer reagents. After *in vitro* digestion samples containing indigestible proteins were freeze-dried. Proteins in white bean paste before and after digestion were extracted as described for SDS–PAGE analysis (Section 2.9.2). The proteins were determined with the Bradford method [26]. Protein digestibility was determined based on the absorbance of samples at 595 nm before and after *in vitro* digestion, measured using a UV-Vis spectrophotometer (UV 1601, Shimadzu Co., Ltd., Kyoto, Japan). Results were expressed as a relative protein digestibility (RPD), compared to digestibility of the control sample (which digestibility was 100%).

The RPD (%) was calculated according to the following formula:RPD (%) = [(AAD_S_/ABD_S_)/(AAD_C_/ABD_C_)] × 100%(3)
where: RPD (%)—relative *in vitro* digestibility of proteins, ABD—absorbance of the sample before *in vitro* digestion, AAD—absorbance of the sample after *in vitro* digestion, C (subscript)—control sample (WBP without phenolics), S (subscript)—studied sample (WBP with phenolics).

### 2.9. Analysis of Protein–Phenolics Interactions

#### 2.9.1. Size Exclusion–High–Performance Liquid Chromatography (SE–HPLC)

Protein extraction for SE–HPLC was carried out as follows: freeze-dried paste samples (100 mg) were suspended in 1 mL 100 mM sodium phosphate buffer (pH 7), vortexed for 1 min and mixed in a rotator for 30 min at room temperature. The samples were centrifuged (MPW 350R, MPW Med. Instruments, Warsaw, Poland) at 9000× *g* for 10 min.

SE–HPLC procedure was performed according to Sęczyk et al. [18]. Samples were analyzed with a Varian ProStar HPLC system (Varian, Palo Alto, CA, USA) equipped with a ProStar diode array detector (DAD). The analytical column was a 600 mm × 7.5 mm COSMOSIL 5-Diol-300-II Packed Column (Nacalai Tesque, Kyoto, Japan). The injection volume of the sample was 50 μL. The proteins were eluted isocratically at 30 °C using a PBS buffer (pH 7.4) with a flow rate of 1 mL/min. The acquisition wavelength was set at 280 nm.

#### 2.9.2. Polyacrylamide Gel Electrophoresis (PAGE)

Protein samples were analyzed by SDS–PAGE according to Laemmli (1970) procedure [27] as well as by its non–denaturing modification–blue native PAGE (BN-PAGE) [28].

For SDS–PAGE proteins were extracted according to the following procedure: 50 mg of freeze-dried white bean paste was suspended 1 mL 1X Laemmli sample buffer (31.5 mM Tris–HCl, pH 6.8, 10% glycerol 1% SDS, 0.005% Bromophenol Blue, 355 mM 2-mercaptoethanol) (BioRad, Hercules, CA, USA) and mixed at rotator for 30 min. Then, samples were boiled for 5 min and all samples were centrifuged (MPW 350R, MPW Med. Instruments, Warsaw, Poland) at 9000× *g* for 10 min to obtain the supernatants. For BN-PAGE, samples were prepared similarly; however without boiling, using SDS (denaturing agent) and 2-mercaptoethanol (reducing agent).

For SDS-PAGE 10 µL and BN-PAGE, 20 µL of protein samples were loaded into each lane. The electrophoretic separation and staining were carried out as described previously [18].

### 2.10. Theoretical Approach

#### 2.10.1. Estimation of the Predicted Phenolic Content and Antioxidant Activity

Estimation of the predicted value (PV) for phenolic content or antioxidant activity allows showing the theoretical phenolic content or antioxidant activity in case of absence of interaction between phenolic compounds and food matrix.

Predicted values (PV) for phenolic content were calculated as follows:PV_P_ = EV_p_(4)
where: PV_P_—the predicted value for phenolic content; EV_p_—experimental phenolic content for phenolic samples without food matrix.

Predicted values for antioxidant activity were calculated as follows:PV_A_ = EV_c_ + EV_p_(5)
where: PV_A_—predicted value for antioxidant activity; EV_c_—the experimental antioxidant activity of control paste (without phenolic compounds), EV_p_—experimental antioxidant activity of phenolic sample without food matrix.

#### 2.10.2. Determination of Combination Index

In this study, the concept of combination index (CI) for the theoretical basis simulation of synergism and antagonism in drug combination studies, proposed in the studies of Chou [29], was applied for the determination of the effect of phenolics–food matrix interactions.

The combination index was calculated as follows:CI = PV/EV(6)
where: PV—predicted value for phenolic content or antioxidant activity; EV—experimental value for phenolic content or antioxidant activity.

CI = 1 indicates the lack of an influence of the phenolics–food matrix interactions on the phenolics content or antioxidant activity (PV = EV), CI > 1 indicates a negative effect of the phenolics–food matrix interactions on the phenolics content or antioxidant activity (PV > EV), and CI < 1 indicates a positive effect of the phenolics–food matrix interactions on the phenolics content or antioxidant activity (PV < EV).

### 2.11. Statistical Analysis

All experiments were carried out in triplicate. The data were subjected to a one–way analysis of variance (ANOVA). Tukey’s test was used to compare the means of the different variables. α values = 0.05 were regarded as statistically significant.

## 3. Results

The effect of food matrix on the phenolics content and *in vitro* bioaccessibility in white bean paste and was shown in Table 1.

To evaluate the effect of white bean food matrix on the free (unbound) phenolics level in the native conditions (before *in vitro* digestion—BD) as well as after *in vitro* digestion (AD), predicted values (PV), and combination indexes (CI) were estimated. In the case of HPLC determination of phenolic compounds in fortified white bean paste, PV has demonstrated the measured phenolics levels of samples without food matrix; thus, they are not equal (but close) to the introduced phenolic amount (10 mg per 1 g DW) due to insignificant effect of paste preparation conditions on phenolics stability. The difference between PV and EV shows the number of phenolic compounds bound to the food matrix, and consequently the affinity of phenolics to the food matrix, which was reflected in the CI values. For BD samples the highest difference between EV (5.44 mg/g DW) and PV (9.97) was determined for CGA (CI = 1.83). The least significant effect of the food matrix on free phenolic abundance was observed for GA (CI = 1.02). The simulated digestion process affects adversely the phenolics stability–for each studied compound PV for AD were lower than for BD. In the case of AD, the highest CI was obtained for Q (CI = 1.68). Furthermore, Q was characterized by the lowest bioaccessibility (BA = 45.4%). Contrary, low affinity to food matrix (CI close to 1), both before (CI = 1.02) and after *in vitro* digestion (CI = 0.98), high stability during processing, and the highest bioaccessibility among all studied compounds (96.0%) was observed for GA. The results also show that the simulated digestion process, compared to the native conditions (BD), can affect positively or negatively on the phenolics–food matrix interactions, depending on the type of introduced phenolic compound. *In vitro* digestion resulted in the increase of affinity of phenolic compounds to white bean components (AD CI > BD CI) for FA and CAT. The opposite effect (AD CI < BD CI) was found for other compounds; however for GA and Q only slight differences were found. For example, CI for a sample of CGA after digestion (C = 1.54) was lower than before digestion (CI = 1.83), which means that during digestion phenolic molecules were released from the food matrix.

The effect of phenolics–food matrix interactions on the antioxidant activity of fortified pasta was presented in Table 2. Results have shown that antioxidant activity varied considerably depending on the applied phenolic compound and the measured parameter. Furthermore, *in vitro* digestion process affected the obtained results. Generally, GA and CAT samples were characterized by higher antioxidant properties compared with other samples, both before and after *in vitro* digestion. Similar to the results obtained for the HPLC analysis, antioxidant activity of phenolic compounds, especially for CGA, Q, and CAT was adversely affected by interactions with the food matrix (EV < PV; CI > 1). Taking into account antioxidant activity the highest “masking” effects of food matrix were found for Q samples, in the case of ABTS antiradical activity determination for samples before digestion (CI = 2.04) and for reducing power after digestion (CI = 1.46). On the other hand, antioxidant parameters for GA samples were only slightly altered by the food matrix both before and after digestion (CI about 1). Moreover, it was found that food matrix effects, both for BD and AD samples, were less significant for chelating power and hydroxyl radical scavenging activity compared to the other parameters.

The results in Table 3 show nutrient contents and *in vitro* digestibility of fortified white bean paste.

The phenolics addition did not have a statistically significant influence on the total dietary fiber content and the total starch content. The determined total dietary fiber content varied from 121.0 mg/g (FA) to 126.5 mg/g (Q) and total starch content from 471.9 mg/g (CAT) to 471.9 mg/g (C). Phenolic compounds significantly reduced the digestible starch content, and consequently, starch digestibility. The most negative impact was observed for CAT (STD 47.8%). Furthermore, phenolic compounds significantly affected the relative protein digestibility (RPD). Compared to the C sample (RPD = 100%), CGA improve RPD by 9.2% (RPD = 109.2%). Other phenolic compounds decreased RPD; however, a statistically significant negative effect was found for CAT, Q and GA. CAT sample was characterized by the lowest RPD among other samples–a decrease of 22.3% in protein digestibility was noticed.

The phenolics–protein interaction was visualized using the SDS–PAGE and BN-PAGE techniques and are presented in Figure 1.

Except CGA and CAT, results from the electrophoretic analysis showed the only slight influence of the most studied phenolics on the proteins separation patterns compared to control. For both CGA and CAT an increase in the intensity of bands at the bottom of resolving gels corresponded to high molecular complexes (HMC) of proteins was observed. Furthermore, SDS-PAGE analysis of CGA and CAT samples, shown higher intensity and bands smearing for proteins corresponding with molecular masses higher than 250 kDa (>250). In addition, for CAT, significantly less intensity of bands corresponding to proteins with molecular mass at the range of 15–20 kDa was found. For BN-PAGE, an increase of intensity of B1 for CAT as well as a slight shift B2 towards the top of the gel for CAT and CGA were noticed.

SE-HPLC profiles of white beans proteins affected by phenolic compounds were presented in Figure 2.

Results show that most of the applied phenolic compounds increased peaks area and total area under the curve of the chromatograms. Compared to the control sample, the least significant effect on the protein profile was observed for FA (Figure 2A) and A (Figure 2C). Similarly to electrophoretic analysis, the most prominent changes were observed for samples enriched with CGA (Figure 2B) and CAT (Figure 2C). The main effect of CGA and CAT addition to white bean paste was reflected in a significant increase of the area of the peak at range of retention time (RT) from 10 to 24 min. What is more, for CAT sample peaks overlapping (RT 12.7 min) slight shift of peaks toward to lower RT (and consequently to higher molecular size) for peaks corresponded to RT 16.2 and 17.8 were observed.

## 4. Discussion

Polyphenols are regarded as valuable phytochemicals characterized by a wide range of pro-health properties. However, it was found that phenolic compounds can interact with the food matrix components such as lipids, carbohydrates (e.g., starch and dietary fiber) and proteins causing the important nutritional and nutraceutical changes in fortified products [5,8,9,10].

It was described that phenolics can interact with food macromolecules mainly reversibly by weak non–covalent forces (hydrogen, hydrophobic, and van der Waals bonds); however, in the case of proteins, irreversible interactions stabilized by covalent bonds are also possible. The affinity of phenolic compounds to food matrix components and interactions strength depends on many external (e.g., temperature, pH, ionic strength) and internal factors (e.g., concentration, type, structure, and physicochemical properties of polyphenols and food matrix components) [8,9,11,30].

As shown in Table 1, interactions of phenolic compounds with white bean components, in most cases, resulted in a significant reduction of the free phenolic level (unbound with food matrix) in white bean paste. Application and estimation of PV and CI allow describing affinity of analyzed phenolic compounds to a food matrix, both in the “native” conditions as well as after the simulated digestion process. It was found that *in vitro* digestion process decreases the bioaccessible phenolic levels. However, comparison CI for samples before and after digestion indicated that this process can significantly affect positively or negatively the phenolics–food matrix interactions, which resulted in an increase or a decrease in phenolics binding, respectively. This finding can be explained by the dualistic effect of the digestion process on the phenolic interactions. Enzymatic degradation and physicochemical modification of the food matrix can lead to a release of the reversely bound phenolic compounds from existing binding sites in the native food matrix (CI AD < CI BD). On the other hand, the food matrix decomposition can uncover “new” binding sites for polyphenols and increase affinity to the food matrix (CI AD > CI BD) [8,13,14,17]. It is also suggested that beyond the negative effect of phenolic interactions with food matrix related to a decrease of phenolics bioaccessibility, interactions may have a positive effect—they contribute to the delivery of phenolic compounds and improve oxidative stability through the gastrointestinal passage [9,14,17,31].

Antioxidant activity is one of the best-known activities of phenolic compounds. It was noticed that phenolics–food matrix interactions, adversely affected the antioxidant activity, in general (Table 2). This effect can be a result of masking antioxidant properties by binding phenolic compounds to the food matrix as well as other more complex factors such as interactions of phenolic compounds with low molecular components of the food matrix e.g., other phenolics, vitamins, and minerals [32]. Previous studies demonstrated that non–covalent interactions of phenolics with other molecules of the surrounding medium can influence their activity by the modulation of reactivity with radical species [33,34].

Furthermore, it was reported that various food matrices with different compositions and physicochemical properties can have different effects on the bioaccessibility and antioxidant activity of phenolic compounds [35,36,37]. Thus, an appropriate combination of food vehicles and phenolic supplements may limit the occurrence of interactions and consequently improve the bioaccessibility of polyphenols. Additionally, except of the composition of the food matrix and the dose of phenolic compounds, phenolics bioaccessibility and antioxidant activity in the food matrix can be improved by adequate processing regarding their stability and modification of food matrix properties towards increased release during digestion [36,38,39,40]. It should be mentioned that the bioaccessibility of phenolic compounds in in vivo conditions can be altered by gut microbiota via metabolic conversion to corresponding derivatives, on the one hand, and via food matrix decomposition by microbial enzymes like carbohydrates (e.g., cellulases, pectinases, glycosidases) and proteases [41], on the other hand. However, the most extensive microbial fermentation occurs at the end fragment of the digestive tract i.e., in the colon, whereas absorption of ingested substances takes place mainly in upper fragments–in the gastric and intestines [2]. Therefore, *in vitro* models can provide valuable basic information about the bioaccessibility of polyphenols, but human studies are still the “gold standard” providing more specific results [13].

Another aspect considered in these studies was the effect of phenolics on the nutritional properties of the fortified product. Some studies show that some phenolic, especially high–molecular compounds, are characterized by high affinity to carbohydrates such as dietary fiber components (e.g., cellulose and pectin) and starch [10,11,42,43]. Due to the possible increase amount of dietary fiber as a consequence of binding polyphenols, its level in fortified paste was determined. However, there was no significant influence between control and fortified samples (Table 3). It may be speculated that was caused by weak strength of interactions and phenolics release from food matrix at conditions during dietary fiber isolation (i.e., ethanol precipitation and washing of dietary fiber).

Moreover, it was found that the phenolics combination with the food matrix modified the starch and protein digestibility under simulated digestion conditions (Table 3). One of the effects of interactions are modifications of the structure and physicochemical properties of macromolecules. Modifications of the properties of proteins and starch can change their susceptibility to enzymatic hydrolysis [8,9,10,11,44].

Due to the biological significance, complex structure, and a wide variety of mechanisms involved in interactions, phenolic–proteins interactions are the most extensively described. For example, protein–phenolics interactions may modify the chemical structure of proteins by covalent bonds between reactive groups of proteins (–NH_2_, –SH, tryptophan residues) and phenolics (–OH) and causing changes in secondary and tertiary as a consonance both of covalent or non–covalent interactions, which results in modification the proteins thermal stability, solubility, hydrophobicity as well as the above-mentioned susceptibility to digestion [9,18,30,45]. It is suggested that structural and physicochemical changes, in general, affect adversely on the protein digestibility, however, positive effects were also described [9,30]. Similarly to obtained results (Table 3), the positive effect of chlorogenic acid on the hydrolysis degree was observed in the case of interactions with soy proteins [45].

In this study, to determine the consequences of phenolic–protein interactions chromatographic and electrophoretic techniques were applied. The most noticeable effect was found for chlorogenic acid and catechin, which correspond to results obtained for *in vitro* digestibility.

Modification of protein profiles can be an effect of intra– and intermolecular interactions of phenolic compounds and proteins [18,30]. Intermolecular cross-linking of proteins leads to the formation of complexes with higher molecular weights which were observed for HMC (SDS-PAGE, BN-PAGE) and >250 kDa fractions (SDS-PAGE) (Figure 1). On the other hand, it may be speculated that a decrease in the intensity of bands at range 15–20 kDa is a result of the participation of these subfractions in complexes formation (Figure 1). Additionally, it seems possible that phenolic–protein interactions, besides structural modifications, can also modify a charge of complexes, which cause changes in the electrophoretic mobility of proteins. Compared to SDS-PAGE, in which proteins mobility depends mainly on the molecular weight of proteins, in native conditions of separation electrophoretic mobility depends on the charge-to-mass ratio, physical shape, and size of the proteins [18,28]. The stability of phenolic–protein complexes in the presence of denaturing and reducing agents such as SDS and 2–mercaptoethanol—which break hydrogen bonds, hydrophobic interactions, and disulfide bonds, suggesting that observed complexes were stabilized by covalent bonds [18]. However, in the case of BN-PAGE and SE–HPLC, protein profiles could have been caused by covalent and non–covalent interactions.

SE–HPLC takes place under more mild chemical behaviour than SDS–PAGE and BN-PAGE; thus, it allows the analysis of phenolic protein complexes stabilized by weaker bonds; however, this method is limited only to soluble proteins in the separation conditions. The increase in the area of peaks (Figure 2) may be a result of proteins association and intramolecular interactions of phenolic compounds. The binding of phenolic compounds to proteins increase UV light absorption (at 280 nm) of complexes, thus higher analytical signals were obtained. Peaks overlapping and shifting toward higher molecular sizes can be a result of proteins association as well as increase protein size by “coating” protein molecules with polyphenols (intramolecular interactions) [18].

Furthermore, it should be noticed that beyond physicochemical and structural changes in the food matrix affected by interactions with phenolic compounds, nutrients digestibility depends also on the phenolics interactions with digestive enzymes [8,9,46]. However, our previous studies concerning the effects of phenolic interactions with white bean proteins and proteolytic enzymes (pepsin, trypsin, chymotrypsin) demonstrated that in most cases interactions with digested proteins have a more significant effect on the hydrolysis degree than interactions with digestive enzymes [47].

## 5. Conclusions

The results demonstrate that phenolic–food matrix interactions affect the *in vitro* bioaccessibility and antioxidant activity of phenolic compounds, mainly in a negative manner. The determination of the combination index allows estimating the affinity of phenolic compounds to the food matrix and the magnitude of the food–matrix effects. Comparison of combination indexes for samples before and after *in vitro* digestion allows demonstrating that phenolic–food matrix interactions change during simulated digestion. In most cases, phenolic compounds negatively affected *in vitro* digestibility of starch and protein. Furthermore, changes in protein electrophoretic and chromatographic profiles caused by interactions with phenolics were noticed. The observed effects of phenolic–food matrix interactions are strictly dependent on the applied phenolic compound, which indicates the complex nature of interactions and individual affinity of phenolic compounds to food matrix components. To summarize, phenolic–food matrix interactions are an important factor affecting the nutraceutical and nutritional potential of fortified products and should be taken into consideration during their design and assessment, however, due to the complexity of interactions it is a long term challenge for food science.

## Figures and Tables

**Figure 1 antioxidants-10-01825-f001:**
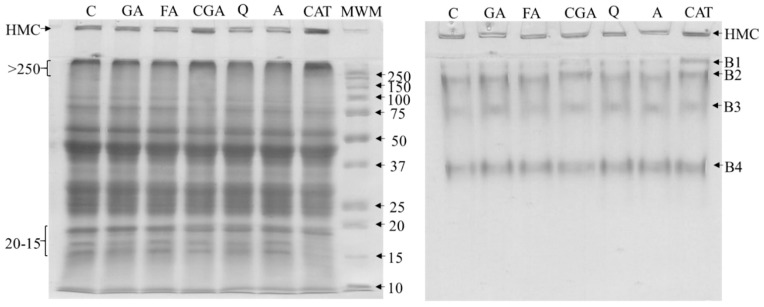
SDS–PAGE (left side) and BN-PAGE (right side) electrophoretic profiles of white bean proteins. Abbreviations: GA—gallic acid, FA–ferulic acid, GGA—chlorogenic acid, Q—quercetin, A—apigenin, CAT—catechin; HMC—high molecular complexes, MWM—molecular weight markers (weight expressed in kDa), B1–B5—main bands in BN-PAGE.

**Figure 2 antioxidants-10-01825-f002:**
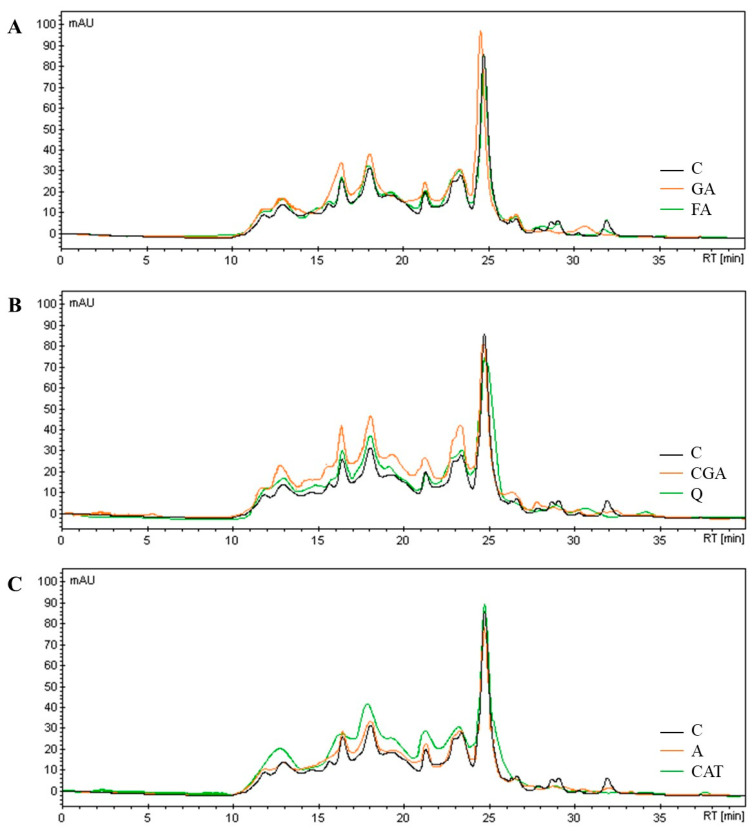
Effect of phenolic addition on SE–HPLC profile of white bean proteins. Abbreviations: C—control, GA—gallic acid, FA—ferulic acid, GGA—chlorogenic acid, Q—quercetin, A—apigenin, CAT—catechin; RT—retention time.

**Table 1 antioxidants-10-01825-t001:** Effect of food matrix on phenolic content and *in vitro* bioaccessibility.

Sample		GA	FA	CGA	Q	A	CAT
BD	EV	9.82 ± 0.22 ^a^	8.38 ± 0.16 ^c^	5.44 ± 0.08 ^e^	5.62 ± 0.15 ^e^	9.28 ± 0.18 ^b^	7.94 ± 0.21 ^d^
	PV	9.97	10.01	9.97	9.96	10.03	9.96
	CI	1.02	1.19	1.83	1.77	1.08	1.25
AD	EV	9.60 ± 0.34 ^a^	6.82 ± 0.20 ^c^	5.84 ± 0.22 ^d^	4.54 ± 0.13 ^f^	8.58 ± 0.21 ^b^	5.38 ± 0.18 ^e^
	PV	9.42	8.52	9.02	7.62	8.46	8.52
	CI	0.98	1.25	1.54	1.68	0.99	1.58
	BA (%)	96.0 ± 1.2 ^a^	68.2 ± 1.9 ^c^	58.4 ± 2.9 ^d^	45.4 ± 2.7 ^e^	85.8 ± 2.0 ^b^	53.8 ± 2.7 ^d^

Means (±SD, *n* = 3) followed by different lowercase letters (superscript) in rows are significantly different at α = 0.05. Abbreviations: BD—samples before *in vitro* digestion, AD—samples after *in vitro* digestion; GA—gallic acid, FA—ferulic acid, GGA—chlorogenic acid, Q—quercetin, A—apigenin, CAT—catechin; EV—experimental value, PV—predicted value, CI—combination index.

**Table 2 antioxidants-10-01825-t002:** Effect of food matrix on antioxidant activity of fortified white bean paste.

Sample			GA	FA	CGA	Q	A	CAT
ABTS (mg TE∙g^−1^)	BD	EV	52.1 ± 0.6 ^a^	43.9 ± 0.6 ^b^	9.3 ± 0.8 ^d^	7.9 ± 0.8 ^e^	9.1 ± 0.4 ^d^	23.0 ± 0.7 ^c^
		PV	51.73	49.41	12.75	16.26	9.51	29.36
		CI	0.99	1.12	1.37	2.04	1.04	1.28
	AD	EV	59.9 ± 1.4 ^a^	39.9 ± 0.7 ^b^	15.9 ± 1.1 ^d^	14.3 ± 0.4 ^e^	15.1 ± 0.4 ^d^	24.2 ± 1.5 ^c^
		PV	61.75	47.10	18.83	18.20	15.15	32.57
		CI	1.03	1.18	1.18	1.27	1.0	1.35
	BA (%)		86.9 ± 1.9 ^b^	110.0 ± 2.9 ^a^	58.4 ± 2.7 ^d^	56.0 ± 2.5 ^d^	60.4 ± 3.3 ^d^	95.1 ± 2.8 ^c^
RP (mg TE∙g^−1^)	BD	EV	15.4 ± 0.5 ^a^	8.5 ± 0.3 ^c,d^	9.0 ± 0.4 ^c^	8.1 ± 0.4 ^d^	6.7 ± 0.9 ^e^	10.0 ± 0.8 ^b^
		PV	16.15	9.46	11.69	13.95	8.05	13.68
		CI	1.05	1.11	1.3	1.72	1.2	1.36
	AD	EV	21.5 ± 0.7 ^a^	8.1 ± 0.4 ^c^	8.7 ± 0.4 ^c^	10.2 ± 0.8 ^b^	6.8 ± 0.8 ^d^	10.7 ± 0.3 ^b^
		PV	20.92	9.02	10.34	14.88	7.78	12.79
		CI	0.97	1.11	1.15	1.46	1.14	1.2
	BA (%)		71.5 ± 3.1 ^d^	104.8 ± 2.5 ^a^	96.7 ± 2.9 ^a,b^	79.1 ± 1.9 ^c^	98.0 ± 3.5 ^b^	94.1 ± 2.5 ^b^
CHEL (mg EDTA∙g^−1^)	BD	EV	0.92 ± 0.01 ^b^	0.96 ± 0.01 ^a^	0.91 ± 0.02 ^c^	0.95 ± 0.01 ^a^	0.85 ± 0.02 ^c^	0.92 ± 0.01 ^b^
		PV	0.94	1.06	1.03	1.14	1.07	0.97
		CI	1.03	1.11	1.13	1.2	1.25	1.06
	AD	EV	0.91 ± 0.01 ^b^	1.01 ± 0.02 ^a^	0.88 ± 0.01 ^c^	0.83 ± 0.02 ^d^	0.81 ± 0.01 ^e^	0.89 ± 0.01 ^c^
		PV	0.92	1.09	0.92	1.05	0.96	0.99
		CI	1.01	1.09	1.05	1.26	1.19	1.11
	BA (%)		100.5 ± 1.9 ^c^	94.9 ± 3.0 ^d^	104.2 ± 2.5 ^b^	114.4 ± 2.8 ^a^	105.6 ± 3.0 ^b^	103.4 ± 2.1 ^b,c^
^•^OH (mg TE∙g^−1^)	BD	EV	166.1 ± 2.5 ^a^	119.7 ± 2.1 ^c^	152.4 ± 4.3 ^b^	60.3 ± 3.4 ^e^	70.0 ± 3.3 ^d^	151.8 ± 4.2 ^b^
		PV	168.49	129.45	156.4	61.93	70.04	169.35
		CI	1.01	1.08	1.03	1.03	1.00	1.12
	AD	EV	161.3 ± 3.8 ^a^	129.8 ± 8.5 ^b^	156.7 ± 3.4 ^a^	61.6 ± 2.6 ^d^	70.1 ± 3.9 ^c^	162.5 ± 3.3 ^a^
		PV	160.39	132.65	158.96	63.98	72.25	169.59
		CI	0.99	1.02	1.01	1.04	1.03	1.04
	BA (%)		103.0 ± 2.7 ^a^	92.3 ± 2.0 ^c^	97.2 ± 2.7 ^b^	98.0 ± 2.3 ^a,b^	99.8 ± 2.9 ^a,b^	93.4 ± 3.2 ^b,c^

Means (±SD, *n* = 3) followed by different lowercase letters (superscript) in rows are significantly different at α = 0.05.Abbreviations: BD—samples before *in vitro* digestion, AD—samples after *in vitro* digestion; GA—gallic acid, FA—ferulic acid, GGA—chlorogenic acid, Q—quercetin, A—apigenin, CAT—catechin; EV—experimental value, PV—predicted value, CI—combination index; ABTS—radical cation scavenging activity, RP—ferric (III) reducing antioxidant power, CHEL—ferric (II) chelating power, ^•^OH—hydroxyl radical scavenging activity.

**Table 3 antioxidants-10-01825-t003:** Effect of phenolic compounds on nutrient contents and *in vitro* digestibility.

Sample	TDF (mg/g)	TS (mg/g)	AS (mg/g)	STD (%)	RPD (%)
C	122.0 ± 5.7 ^a^	478.9 ± 4.9 ^a^	260.9 ± 3.2 ^a^	54.5 ± 1.0 ^a^	100.0 ± 2.3 ^b^
GA	124.0 ± 7.1 ^a^	477.9 ± 4.9 ^a^	243.3 ± 2.3 ^c^	50.9 ± 0.9 ^b^	87.4 ± 3.1 ^c^
FA	121.0 ± 4.2 ^a^	474.6 ± 5.5 ^a^	247.6 ± 1.9 ^b^	52.2 ± 1.0 ^b^	98.6 ± 2.4 ^b^
CGA	124.5 ± 3.5 ^a^	471.9 ± 5.5 ^a^	235.9 ± 2.8 ^d,e^	50.0 ± 0.2 ^c^	109.2 ± 2.8 ^a^
Q	126.5 ± 3.4 ^a^	473.6 ± 6.8 ^a^	238.2 ± 1.4 ^c,d^	50.3 ± 0.4 ^c^	85.4 ± 3.4 ^c,d^
A	122.5 ± 3.1 ^a^	475.2 ± 6.5 ^a^	241.6 ± 2.0 ^c^	50.8 ± 1.0 ^b,c^	95.2 ± 2.5 ^b^
CAT	125.0 ± 5.7 ^a^	476.4 ± 5.0 ^a^	231.9 ± 4.1 ^e^	47.8 ± 0.9 ^d^	78.7 ± 4.0 ^d^

Means (±SD, *n* = 3) followed by different lowercase letters (superscript) in columns are significantly different at α = 0.05. Abbreviations: TDF—total dietary fiber, TS—total starch, AS—available starch, STD—starch digestibility, RPD—relative protein digestibility; GA—gallic acid, FA—ferulic acid, GGA—chlorogenic acid, Q—quercetin, A—apigenin, CAT—catechin.

## Data Availability

Not applicable.

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
