# Peer review of "Influence of Phenolic-Food Matrix Interactions on In Vitro Bioaccessibility of Selected Phenolic Compounds and Nutrients Digestibility in Fortified White Bean Paste"

_antioxidants, 2021, doi:10.3390/antiox10111825_

Round 1

Reviewer 1 Report

In this manuscript authors presented their study that focus on the interactions of phenolic compounds and the components of food matrix, in order to find any significant change in nutritional quality. The results were well  presented and  consistent following different methods.  In the paragraph of discussion authors should add potent solutions that could help maintain the antioxidant activity of phenolic compounds in food matrix. In addition, authors should mention the microbial metabolism that may play a key role in polyphenol bioaccessibility, as well as the different impact that may have different food matrices.

Minor comments:

-In line 93, authors should specify if the amount of 10 mg refer to each phenolic compound or it is the final concentration (authors in this case should mention the quantity for each phenol compound).

-In line 488 there is an extra comma.

Author Response

We would like to thank you very much for your constructive and inspiring comments and remarks. We are convinced that these revisions will improve and clarify the message of the manuscript. We would like to apologize for the mistakes that should not have happened in the final version of the manuscript. Below, we provide a list of detailed answers to the Reviewer 1 questions/comments.

Following Reviewer #1 instructions:

Moderate English changes required

Response: The English language was improved. 

In the paragraph of discussion authors should add potent solutions that could help maintain the antioxidant activity of phenolic compounds in food matrix. In addition, authors should mention the microbial metabolism that may play a key role in polyphenol bioaccessibility, as well as the different impact that may have different food matrices.

Response: The Discussion section was improved according to reviewer suggestions (lines 432-450).

-In line 93, authors should specify if the amount of 10 mg refer to each phenolic compound or it is the final concentration (authors in this case should mention the quantity for each phenol compound).

-In line 488 there is an extra comma.

Response: The manuscript was corrected accordingly.

Reviewer 2 Report

Review for the manuscript entitled Influence of phenolic–food matrix interactions on in vitro bio-accessibility of selected phenolic compounds and nutrients digestibility in fortified white bean paste (antioxidants-1454885)

This manuscript presents an interesting study about the interaction between phenolics and food matrix. The study is well designed and although the description of the methods might be improved, the results and discussions are well presented and documented. Generally, the manuscript is well written. In my opinion, after some minor revisions it can be published in the Antioxidants.

Here are some comments for the Authors:

Materials and Methods: Please provide more details about the apparatus used in experiments (producer, model, city, country). For example:

L83: laboratory mill

L88: water bath

L101, L182, L215, L232: centrifuge

L154-158, L201: spectrophotometer

Other formal suggestions:

L109: with a small modification

L122: please correct into SGFESS

L243, L247, L257: as follows

Author Response

We would like to thank you very much for your constructive and inspiring comments and remarks. We are convinced that these revisions will improve and clarify the message of the manuscript. We would like to apologize for the mistakes that should not have happened in the final version of the manuscript. Below, we provide a list of detailed answers to the Reviewer 2 questions/comments.

Following Reviewer #2 instructions:

Materials and Methods: Please provide more details about the apparatus used in experiments (producer, model, city, country). For example:

L83: laboratory mill

L88: water bath

L101, L182, L215, L232: centrifuge

L154-158, L201: spectrophotometer

Response: More details about the apparatus used in experiments were provided.

L109: with a small modification

L122: please correct into SGFESS

L243, L247, L257: as follows

Response: The manuscript was corrected accordingly.

Reviewer 3 Report

Manuscript "Influence of phenolic – food matrix interactions on in vitro bioaccessibility of selected phenolic compounds and nutrients digestibility in fortified white bean paste" presents very interesting and innovative research. The manuscript is well thought out, research is well planned and the results are clearly presented.

Detailed comments:

Abstract - in this chapter it is good for the authors to provide more research results.

Introduction - chapter written correctly.

The methodology is very clearly written, you can recreate the research on the basis of this chapter.

In tables, you can remove most of the horizontal lines (leave only the separating samples), then it would be easier to read.

Table 2 - standard deviations should have the same number of significant places as the result, the authors should use SD approximations.

Figure 2 - Could the drawings not be colored? It's hard to tell different shades of gray.

The discussion is conducted correctly.

Conclusion is written correctly, it is understandable. 

Author Response

We would like to thank you very much for your constructive and inspiring comments and remarks. We are convinced that these revisions will improve and clarify the message of the manuscript. We would like to apologize for the mistakes that should not have happened in the final version of the manuscript. Below, we provide a list of detailed answers to the Reviewer 3 questions/comments.

Following Reviewer #3 instructions:

Abstract - in this chapter it is good for the authors to provide more research results.

Response: The abstract was improved. Selected research results were added (respecting abstract length limitations).

In tables, you can remove most of the horizontal lines (leave only the separating samples), then it would be easier to read.

Table 2 - standard deviations should have the same number of significant places as the result, the authors should use SD approximations.

Response: The tables were corrected accordingly.

Figure 2 - Could the drawings not be colored? It's hard to tell different shades of gray.

Response: The manuscript was corrected accordingly. Drawings from Figure 2 were replaced by coloured ones.